# Prevalence of Malnutrition in Pediatric Hospitals in Developed and In-Transition Countries: The Impact of Hospital Practices

**DOI:** 10.3390/nu11020236

**Published:** 2019-01-22

**Authors:** Andrea McCarthy, Edgard Delvin, Valerie Marcil, Veronique Belanger, Valerie Marchand, Dana Boctor, Mohsin Rashid, Angela Noble, Bridget Davidson, Veronique Groleau, Schohraya Spahis, Claude Roy, Emile Levy

**Affiliations:** 1Research Centre, CHU Sainte-Justine, Montreal, QC H3T 1C5, Canada; twisty555@hotmail.com (A.M.); valerie.marcil@umontreal.ca (V.M.); v.belanger.7@gmail.com (V.B.); val.marchand@sympatico.ca (V.M.); schohraya.spahis@recherche-ste-justine.qc.ca (S.S.); microbionutrisymposia@gmail.com (C.R.); 2Departments of Nutrition, University of Montreal, Montreal, QC H3C 3J7, Canada; 3Departments of Biochemistry, University of Montreal, Montreal, QC H3C 3J7, Canada; 4Departments of Pediatrics, University of Montreal, Montreal, QC H3C 3J7, Canada; veronique.groleau@umontreal.ca; 5Pediatric Gastroenterology, Alberta Children’s Hospital, University of Calgary, Calgary, AL T2N 1N4, Canada; Dana.Boctor@albertahealthservices.ca; 6Dalhousie University and IWK Health Center, Halifax, NS B3H 1S6, Canada; Mohsin.Rashid@iwk.nshealth.ca (M.R.); Angela.Noble@iwk.nshealth.ca (A.N.); 7Canadian Nutrition Society, Ottawa, ON K1C 6A8, Canada; bdavidson1017@bell.net

**Keywords:** hospital malnutrition, hospital stay length, nutritional screening tools

## Abstract

Presently, undernutrition still goes undetected in pediatric hospitals despite its association with poor clinical outcomes and increased annual hospital costs, thus affecting both the patient and the health care system. The reported prevalence of undernutrition in pediatric patients seeking care or hospitalized varies considerably, ranging from 2.5 to 51%. This disparity is mostly due to the diversity of the origin of populations studied, methods used to detect and assess nutritional status, as well as the lack of consensus for defining pediatric undernutrition. The prevalence among inpatients is likely to be higher than that observed for the community at large, since malnourished children are likely to have a pre-existent disease or to develop medical complications. Meanwhile, growing evidence indicates that the nutritional status of sick children deteriorates during the course of hospitalization. Moreover, the absence of systematic nutritional screening in this environment may lead to an underestimation of this condition. The present review aims to critically discuss studies documenting the prevalence of malnutrition in pediatric hospitals in developed and in-transition countries and identifying hospital practices that may jeopardize the nutritional status of hospitalized children.

## 1. Introduction

Although malnutrition in pediatrics is of concern in low resource settings, this state is also of major worry for hospitalized children in developed as well as in-transition countries [1,2,3,4]. However, causes for malnutrition differ in the two environments. Independently of the income setting, malnutrition is multifactorial. Whereas malnutrition in low-income countries is often, but not solely, attributable to limited access to food and/or medical care, it is often triggered by disease in in-transition countries [5,6]. Of importance, the report of the Global Burden of Disease Study 2013 revealed that protein-energy malnutrition accounted globally for 9.8/100.000 age-standardized deaths in the largest 50 countries for child and adolescent populations. More alarming, when classifying the data according to the level of development, it accounted for 11/100.000 age-standardized deaths in the developing countries and 0.1/100.000 age-standardized deaths in developed countries [7].

Poor nutritional status at admission or worsening of nutritional status during hospitalization is recognized to adversely affect clinical outcomes. Among other systems, it disturbs immune response, thereby causing children to have piteous wound healing with higher risk of infections and complications of their underlying disease [8,9,10,11,12,13]. Furthermore, these adverse effects lead to delayed recovery and prolong hospitalization, thereby increasing the financial burden on the health care system (in-patient day costs, treatments) and limiting hospital bed availability [8,9,10,14,15,16,17,18,19,20,21]. As an example, a Canadian study revealed that malnourished children [evaluated by the Subjective Global Nutrition Assessment (SGNA) scheme], undergoing thoracic or abdominal surgery, were more likely to develop infectious complications while experiencing longer post-operative length of stay compared to well-nourished patients [9]. Also, a prospective observational study that included 44 Netherland pediatric wards [16], showed that individuals with acute malnutrition remained hospitalized 45% longer than well-nourished patients. In 2007, the British Association for Parenteral and Enteral Nutrition reported that disease-related malnutrition (DRM) in UK generated annual costs of more than £ 13 billion [22]. More recently, Freijer et al. conducted a cost-of-illness analysis to evaluate the additional costs that DRM imposes on the Netherlands health care system. They estimated that annual additional medical costs related to acute DRM for hospitalized children amounted to 51 million € with a prevalence of 12% [23].

In comparison to adults, children are particularly vulnerable to malnutrition, having a lower caloric reserve and higher nutritional requirements per unit of body weight, to account for growth [24,25]. When factoring in the impact of disease or illness that contributes to increased nutrient requirements, malnutrition may, on long-term, impact the growth and cognitive development trajectory [3,26,27]. It follows that early identification of malnourished children or children who are potentially at risk for malnutrition is key to preventing debilitating sequels.

The objectives of the present review are to provide a short historical account, briefly describe the tools developed for nutritional screening and assessment, present an overview on the prevalence of pediatric malnutrition in patients seeking care and/or hospitalized, and document the compliance of the published guidelines in pediatric hospitals in developed and in-transition countries. We have selected examples of countries that are classified as developed economies and economies in transition according to the definition used by World Economic Situation and Prospects and prepared by the Development Policy and analysis Division of the Department of Economic and Social Affairs of the United Nations Secretariat [28].

## 2. Methods

This review encompasses guidelines and clinical studies published from 1995 to 2018, which address the screening, assessment and management of malnutrition in hospitalized children in industrialized countries. The PubMed database was searched. The following generic search terms were used: malnutrition, pediatric hospital, assessment and prevalence with the following filters: clinical studies, human, publication dates from 1995 to 2018, language: French and English, and aged from birth to 18 years old.

## 3. Malnutrition: A Broad Concept

The generic term malnutrition encompasses deficient, excessive or imbalanced intake of a variety of nutrients jeopardizing the health status. It can be causal or consequential. The present review essentially addresses acute and chronic undernutrition. In the context of this review, malnutrition is synonym of undernutrition and results from disease-related deprivation or malabsorption of nutrients, leading to altered body composition [29]. It is quite distinct from disease-free malnutrition, which is related to hunger-, socioeconomic- and psychologic-related conditions and does not include failure to thrive, defined as a deficient weight gain and related to chronic conditions.

As of today, there is still no consensus on the best definition of pediatric malnutrition. This lack of agreement, accounting partly for the disparity in the reported malnutrition prevalence, impacts on child health outcomes as it precludes adequately identifying children at risk of malnutrition/undernutrition. Assessing the nutritional status in children with moderate, acute malnutrition is particularly problematic as no single indicator can be used alone. Different definitions and classification methods have been used over the years to describe malnutrition based on anthropometric parameters in children. A brief historical account follows.

## 4. Succinct Historical Reference

Gomez et al. [30] in 1956 provided one of the earliest classification systems categorized on percentage of expected weight for age. However, this approach relied solely on weight measurements and had drawbacks. First, it was not always practicable in developing countries where many parents did not know the precise age of their child. Second, many children with short stature were often mistakenly classified as severely malnourished while in fact their body weight was appropriate for their height [31]. The group of Waterlow [32] proposed in 1972 that acute malnutrition be defined independently of age and suggested using weight in relation to height. To our knowledge, they were the first to differentiate acute malnutrition or wasting [described by the weight-for-height (WFH) percentile], from chronic malnutrition or stunting by the height-for-age (HFA) percentile. In 1999, the World Health Organization (WHO), recommended in their manual for the management of severe malnutrition the assessment of nutritional status according to the presence of edema, WFH and HFA z-scores [33]. The WHO classified malnutrition as moderate (absence of edema + z-scores between −2 and −3) or severe (presence of edema + z-scores <−3). Mild malnutrition was not included in the classification [33].

In 2013, the Pediatric Malnutrition Definitions Working Group (PMDWG), commissioned by the American Society for Parenteral and Enteral Nutrition (ASPEN), defined malnutrition (undernutrition) in their thorough report as “an imbalance between nutrient requirements and intake that results in cumulative deficits of energy, protein or micronutrients that may negatively affect growth, development and other relevant outcomes” [31]. The expert group further sub-classified it on its etiology: secondary to disease/injury or environmental/behavioral factors or both. The PMDWG identified the following key concepts that have to be considered when defining pediatric malnutrition: anthropometric variables, chronicity and severity of malnutrition, etiology and pathogenesis of nutrient imbalance, and functional outcomes (Figure 1). In the proposed practical scheme, they classified malnutrition as acute or chronic with 3 months as the threshold for the latter, illness-related or not for the etiology, presence or absence of an inflammatory state and pathogenic mechanisms resulting in nutrient intake/absorption less than requirements. Furthermore, endorsing the WHO guidelines, the ASPEN-working group recommended that the results of anthropometric measurements be expressed as z-scores. In this context, their 2014 consensus statement defined mild malnutrition as a z-score −1 to −1.9, moderate malnutrition a z-score between −2 and −2.9 and severe malnutrition a z-score <−3. This American group proposed that malnutrition indicators namely WFH or Body Mass Index (BMI) or HFA should be used with WHO child growth standards from birth to 2 years old and CDC growth charts for children ages 2−20 years [34]. In 2017, a joint multidisciplinary task force of the ASPEN and of the Society of Critical Care Medicine published guidelines for best practice in nutrition therapy targeting critically ill patients admitted in a Pediatric Intensive Care Unit (PICU) [35]. The group recommended that upon admission in the PICU patients undergo a thorough nutritional assessment with weekly re-assessment throughout the stay. They further recommended that BMI or WFA z-scores be used when <2 years old and that head circumference be documented when less than 36 months old. However, they did not specify the thresholds to be used. The comprehensive classification scheme and the accompanying recommendations should help clarify the attitude of the clinicians toward this important public health issue.

In the following sections, we will describe the different clinical tools developed for assessing malnutrition and verify whether these tools have been applied for children admitted to hospitals in developed and in-transition countries. Finally, we will attempt to examine whether hospital practice related to nutritional support have improved since 1995, when the first nutritional management tools were reported.

## 5. Nutrition Screening Tools

Malnutrition at admission as well as nutritional status deterioration during hospital stay may lead to adverse events in children. However, malnutrition evaluation and classification are part of nutritional assessment, a complex and time-consuming process that must consider features other than growth indicators. Therefore, clinical tools were developed to identify patients who may benefit the most of a full nutritional assessment, which is the goal of malnutrition screening [36,37]. Seven malnutrition screening tools for hospitalized children have chronologically been proposed since 1995: The Pediatric Nutritional Risk Score (PNRS) in 2000, the Screening Tool for Risk on Nutritional status and Growth (STRONG_kids_) in 2010, the Pediatric Yorkhill Malnutrition Screening (PYMS) in 2011, the Screening Tool for the Assessment of Malnutrition in Pediatrics (STAMP) in 2012. Moreover, in 2015, new tools were introduced to serve the general pediatric population: The Pediatric Digital Scaled Malnutrition Risk Screening Tool (PeDiSMART) and the Pediatric Nutrition Screening Tool (PNST), as well as the modified STAMP: The Pediatric Malnutrition Screening Tool (PMST) in 2016. They are, however, not yet fully validated [36]. These tools aimed to detect patients with early signs of nutritional status alterations and to classify children according to their risk of developing nutritional and medical complications during hospitalization (low, moderate or high). They use a specific scoring system based on patient characteristics and medical condition [36,38]. Table 1 summarizes their characteristics. Reilly et al. [39] in 1995 developed the NRS with the intent of covering all age groups. The age of the 153 patients included into their study ranged between 8.5 months to 93 years. All parameters were the same for the complete range except for BMI that was omitted for the pediatric age group. Intriguingly, there was no mention of the WHO guidelines that were established well before their publication.

Five years later, Sermet-Gaudelus et al. in 2000 [42] published a nutritional status assessment tool adapted to pediatrics (PNRS). It was developed in a study comprising 296 children (mean age: 15 months), from mild undernutrition to severe undernutrition, and based on Percentage Ideal Body Weight. The following nutritional risk factors were appraised: food intake, difficulty retaining food, pain and ability to eat. As no validated classification system for pathologic condition was available, they derived their own system for classifying pathologic conditions and mainly based on those of the American Academy of Pediatrics and the American Dietetic Association [47]. Then, they developed a nutritional risk score that represented the probability of losing >2% of the reference weight as a function of the number of risk factors. The authors concluded that poor food intake, pain and disease severity were the 3 most predictive factors of weight loss during hospitalization.

Hulst et al. [41] in 2010 argued that tools published previously were in favor of nutritional assessment than nutritional risk evaluation. They hence reported on their attempt of developing an easily clinically applicable “STRONG_kids_” tool standing for Screening Tool for Risk on Nutritional status and Growth. It consisted of 4 elements: subjective clinical evaluation (decreases sub-cutaneous fat, muscle mass wasting, hollow face), high-risk disease or expected major surgery, nutritional intake excessive diarrhea, vomiting, reduced food intake, pre-existing dietary intervention, inability to consume adequate quantity of food due to pain) and weight loss/no weight gain each attributed a score of 1 or 2 points with a maximum score of 5. Objective elements were weight for and length for height expressed as WFH and HFA standard deviation-scores. Patients scored 1 to 3 had similar WFH, they were regrouped as moderate risk and those with higher score (4 to 5) were classified as high risk. The length-of-hospital stay as one of the outcomes was directly related to the global score. However, no data was reported on the relation between the nutritional risk score and the readmission rate, an important morbidity index.

Gerasimidis et al. [45] in 2010, reported the evaluation of PYMS, based on the ESPEN guidelines published in 2003 [48]. It assessed four elements: BMI, history of weight loss, dietary intake and predicted effect of the underlying pathology on nutritional status, each with a score ranging from 0 to 2. Patients with a total score of 2 were judged as high risk of malnutrition. They also included body composition as a discriminant by either measuring fat and muscle mass by impedance (children >5 years old) or mid-upper arm circumference & triceps/subscapular skinfold thickness in those <5 years. They compared this tool with the full nutritional assessment and 2 other peer-reviewed assessment tools: SGNA and STAMP. A moderate agreement was observed with the full nutritional assessment, but had a similar sensitivity but higher positive predictive value than the STAMP, and lower specificity and higher sensitivity than the SGNA.

McCarthy et al. [40] developed and evaluated in 2012 STAMP^©^, (Scholl of Biomedical Sciences, University of Ulster, Cromore Rd, Coleraine, UK) a nutrition assessment tool targeting children above 2 years old and adolescents that consisted of 3 elements, namely: anthropometric factors (low percentile WFA, reported weight loss, discrepant weight for height percentile), dietary factors (recent modification of appetite, sub-optimal dietary intake in the recent past) and risk related to the clinical condition. Each of these elements was scored. A global score ≥4 meant high risk of malnutrition. In the context of the evaluation environment, STAMP had a sensitivity of 70%, a specificity of 91%, a positive predictive value of 0.548 and a negative predictive value of 0.948. It however displayed a moderate reliability when compared to the full nutritional assessment and but compared well with the PYMS [40].

PMST, a recent tool available is in fact a minor modification of the STAMP by adding the BMI concept to define obesity (BMI > 98th percentile) or overweight (BMI > 85th percentile) [7]. Finally, Karagiozoglou-Lampoudi et al. proposed PeDiSMART, a computer-based tool to evaluate malnutrition risk of adverse outcome [46]. They observed a significant negative relationship between the PeDiSMART score and HFA, BMI and triceps skinfold thickness z-scores, as well as direct relationship with weight loss during hospitalization. and length of stay. This tool remains yet to be fully validated. They also reported a good correlation with STAMP, PYMS and STRONG_kids_. Interestingly, Huysentruyt et al. [49] in a systematic review aimed at evaluating the accuracy of 4 validated tools (STAMP, PYMS, PNRS, STRONG_kids_) for assessing nutritional risk in hospitalized children in developed countries concluded there were insufficient data to select one tool over another, and that the choice should be related to the availability of resources and dietetic staff.

On the other hand, Secker et al. [9], arguing first that methods of assessing the nutritional status in children based on objective measures are difficult to apply in clinical settings, developed and validated the SGNA in 2007. This assessment tool was essentially based on clinical judgment and Bayesian analysis previously (1982) reported by Baker et al. [50]. The objective of this refined tool was to identify pre-surgical malnutrition and to predict post-surgical nutrition-associated-complications that could lead to prolonged hospital stay and enhance morbidity. The validation study targeted patients aged between 1 month and 18 years and compared the proposed SGNA to objective parameters. The subjective elements included history of current weight and height, dietary intake, frequency and duration of gastrointestinal symptoms, and current and changes in functional capacity. Further to these, physical examination was performed to detect fat and muscle wasting and edema. These parameters were then used to establish a global rating of the patient’s nutritional status (well-nourished, moderately malnourished or severely malnourished). No strict scoring system was used. The objective nutritional markers comprised: length/height (depending on age), weight, % ideal body WFH, BMI-for-age, mid-arm circumference, triceps skinfold thickness, mid-arm muscle area, handgrip strength, serum albumin, transferrin, whole blood hemoglobin and lymphocyte count. The primary outcome was the presence or absence of nutrition-associated complications (NACs) 30 days post-surgery. Other outcomes were the post-operative length of stay, use of non-prophylactic antibiotics and unplanned reoperation or readmission. Under the experimental conditions and clinical setting, the authors showed that the SGNA had a good correlation with the objective measures of the nutritional status, and thus constitutes a clinically valid tool for identifying children at higher risk of NACs and prolonged hospitalization.

In summary, despite the considerable efforts devoted collectively by the health professionals, as of today there is still no reference or a gold standard method for nutritional of hospitalized patients in a pediatric setting. Nevertheless, many national guidelines call for malnutrition risk screening and assessment of patients upon hospital admission and ideally during their stay. However, these tools should be considered for evaluating the nutritional risk and not absolute diagnostic tools.

## 6. Prevalence of Acute Malnutrition

The disparity of the reported prevalence of acute malnutrition in hospitalized children stems from the population studied, clinical settings and tools for defining malnutrition. The sample studies cited in Table 2 exemplify this situation. Although variables aiming at evaluating chronic malnutrition are included in some of the studies, this narrative review mainly targets acute malnutrition. As can be appreciated, almost all studies were conducted in tertiary care facilities. They vary in population sample size studied (43 to 2.4 million), cover a wide age range (1 m–18 y) and utilize different assessment tools and/or variables. In addition, investigators report differently the deviation from normality, using varying % differences from a median, z-scores or centiles. This heterogeneity warrants caution in the interpretation of the data. For instance, using the Waterlow criterion, with a WFH threshold at <80% of median to reflect moderate to severe acute protein-energy malnutrition or wasting, Hendricks et al. [51], Pawellek et al. [52] and Toole et al. [53] established respectively the prevalence of undernutrition at 7.1%, 6.1% and 17.4%. The higher prevalence observed in the last study could partially be explained by the patients being admitted in a critical care facility rather than general pediatric wards. Pawellek et al. [52] reported a higher prevalence (17.2%) for their cohort when assessing malnutrition based on triceps skinfold thickness or when using different WFH cut-offs (<90% of the median: 24.1%); WFH 81–90% of the median: 17.9%). Similarly, Huysentruyt et al. [14] estimated the prevalence of acute malnutrition at 9.0% when based on WFH < −2 SD, 2.4% on %WFH < 80%, 9.8% on BMI < −2 SD and 3.8% on MUAC < −2 SD. Hendriske [54] reported a prevalence of 8% when based on <−2 SD or <5%ile. WFA and 16% when based on <80% of STD WFH. The situation is not much better when specific tools were used. Sermet et al. [42], Hankard [55] and Groleau [56] using the PNRS reported that 44.2%, 26% and 20.2%, respectively, of the children were at risk of being malnourished. In the first two studies, the prevalence of malnutrition was 26% and 12% when children were assessed on the PIBW and BMI respectively. A cross-sectional study of Dogan et al. [57] performed in regional Turkish hospitals, reveals an elevated rate of moderate to severe undernutrition, based either on WFA (36.6%), WFH (27.7%) or BMI < −2 SD (43.4%). Finally, Marteletti et al. [58] in a 1-day cross-sectional survey performed during 3 different seasons in regional and university hospital settings, observed that 11% of the children, aged 2 months to 16 years old, were undernourished. Recently our group [59], using growth parameters (WFA or BMI or WFH or HFA < −2 SD), reported that 19.5% of children admitted in Canadian hospitals were undernourished. More importantly, we also demonstrated that their condition worsened during their stay as the mean WFA Z-score was lower at discharge. Interestingly, the percentage of patients that lost weight during hospitalization was significantly lower in those visited by a dietician.

Larger-size studies would be expected to provide data with less variation. Six studies that included ±1000 patients recruited in different hospital settings, therefore better representing the “true” prevalence for the respective geographical locations have been reported. Wyrik et al. [64] from the US reported a prevalence of malnutrition of 24.5% based on ≤5%ile BMI. However, as the population studied was from emergency departments of tertiary care facilities there is a likelihood of selection bias. Cao et al. [66] from China reported that 9.1% of children were at risk of malnutrition based on the STRONG_kids_ criteria and 14.5% were malnourished based according to <−2 SD BMI in a population selected from tertiary care facility pediatric wards. Chourdakis et al. [60] compared the prevalence of malnutrition risk obtained with PYMS, STAMP and STRONG_kids_ criteria in 1258 children from 14 European general hospital pediatric and surgery wards. The prevalence was 22%, 22% and 10% respectively. Although there was a good agreement between PYMS and STAMP, STRONG_kids_ criteria provided a lower prevalence reflecting again the difference in the outcomes measured. More recently, in a study involving 1994 patients from Italian tertiary care and general pediatric wards and using either indiscriminately BMI or WFH, Lezo et al. [62] reported a prevalence of 13.2%, a value lying close to that reported by Chourdakis et al. [60] when using the STRONG_kids_ criteria. Carvalho et al. [65] in a US nationwide survey, covering 2.14 × 10^6^ children and using the International Classification of Disease (9th Revision) to identify coded diagnoses of pediatric malnutrition based on an etiology-related definition, reported a prevalence of 3.7%, in 2011 compared to 1.9% in 2002, the lowest values reported in the studies listed in Table 2. The authors, however, commented that such a survey might well underestimate the true prevalence of malnutrition. Beser et al. [63] recently reported extremes in the prevalence of malnutrition in a nationwide survey. When PYMS criteria were used for a subset of subjects, they observed that almost 40% of the children were at risk of malnutrition compared to 3.4% when the STRONG_kids_ criteria were considered. They reported intermediate values for BMI (9.5%) and WFA (14.8%). Finally, Sissaoui et al. [67] reported a prevalence of 11.9% of undernutrition based on a WFH < −2 SD in a group of tertiary care patients with widely varying age (1 d–16 y).

In conclusion, these examples amply illustrate the inter- and intra-study difference in the prevalence of malnutrition depending on the variables used, subjective evaluations vs. objective measures, interpretation of the thresholds, and selected severity degrees. Despite this, the severity of the illness, the existence of underlying diseases, and the young age have all been identified as risk factors for malnutrition. Last but not the least, the prevalence estimates also varied according to the population sample. For instance, a higher prevalence of malnutrition would be expected for toddlers and patients below 5 years of age admitted to a PICU, compared to subjects admitted in a general pediatric ward.

It is noteworthy that, 46 years ago, Baker et al. [68] argued that individual measurement tools for assessing the nutritional status had a low predictive value and that combining these measurements into a statistical index did not create a technique with sufficient predictive power to identify high-risk patients in a clinical setting. Although the field has evolved since then, this issue remains a matter of debate. In fact, the more recent studies presented in Table 2 address the negative or positive predictive values of nutritional status assessment tools for outcomes, including Length of Stay for weight loss during hospitalization. Nonetheless, awareness of the seriousness of the malnutrition issue builds support for the importance to develop the most appropriate tool to improve hospital practices and enhance quality of patients’ care. This review underscores the need of implementing nationwide benchmarking programs that would allow early documentation of at-risk patients, estimation of the real prevalence, clinical consequence, and social burden of pediatric malnutrition.

## 7. Hospital Practices

Recognizing the varying prevalence of undernutrition in hospitalized children, the question is whether hospital measures are undertaken to manage this condition. In 1994, Butterworth [69], in his provoking paper “The skeleton in the hospital closet” that attracted a widespread attention of the healthcare community, exposed that malnutrition of hospitalized patients was too often undiagnosed and untreated. He indicated several hospital practices that lead to worsening of the patient’s nutritional status, and strongly advocated the need for practitioners to acknowledge this issue in order to improve nutrition practice in hospitals. While some countries have shown efforts by integrating mandatory guidelines for nutritional assessment in hospitals, others are still far behind despite the consequences for the patient’s health and burden for their respective health care systems.

Malnutrition is multi-factorial. Although the disease state or trauma may deteriorate the nutritional status of children, some hospital practices may exacerbate the condition [49,69,70,71]. As highlighted in Table 3, they consist of failure of documenting upon admission patient’s weight and height, plotting these on inappropriate growth charts, inaccurately measuring anthropometric variables either due to lack of proper equipment or inadequate staffing, failure of documenting nutritional status or reporting nutritional intake due to lack of dietetic referral or inadequate nutritional education and training of hospital staff, providing ill-adapted hospital food. The omission of documenting weight and height upon admission in pediatric wards is concerning as it will result in failing to identify acutely or chronically malnourished children. Additionally, it will prevent health care practitioners evaluate whether the nutritional status is deteriorating during the hospital stay, thereby affecting the patient’s safety. This situation has been extensively documented, as briefly chronologically described below.

Bunting et al. [73] in a prospective study reported that while 83% of the nursing notes recorded weight only 13% of the medical notes did. They also reported that 40% of patients’ files had growth charts present, but for a fair percentage either weight or height were missing. Even more concerning was that only 17% of surgical ward records mentioned weights and heights compared to 65% for the medical ward records.

In a retrospective chart review of US well-child clinics, Chen et al. [74] evaluated the frequency with which clinicians plotted growth measurements and documented growth abnormalities during health maintenance visits. Health care providers failed to plot at least one of height, weight, and/or head circumference measurements in 21% visits and overlooked growth abnormalities in 55% visits. In a nutrition audit of an Australian tertiary care pediatric hospital, O’Connor et al. [75], reported that 73% height measurements were absent on bed charts, and that 12% combined height/weight measurements were missing. They further indicated that inaccuracy in measuring height varied according to the wards where patients were hospitalized, and more importantly that only 18% of undernourished children were referred to dietetic services. A year later, in a 14-day audit of 491 hospital charts in a Canadian tertiary care pediatric hospital, Cummings et al. [76] reported that height/length was recorded in none of the Emergency Department and that growth charts were found in only 23% of the hospital ward records.

Similarly, in a retrospective study, Ramsden et al. [77] reported that whilst only 5% of children had their stature documented in the medical charts, their weight measurements were documented in 95% of the cases. However, a growth chart was present only in 27% of charts and 7% with the contemporary measurements plotted. Milani et al. [78] on their part, demonstrated that prior to introducing nutritional evaluation, only 10–15% of height and weight measures upon admission were plotted on a growth chart and inserted into patients’ medical record, and that the practice improved significantly one year after introducing the tool. These studies show that there is variability in the quality of services provided, even in a tertiary care facility. They also underline that guidelines decreed by International bodies such as WHO, AAP and CDC are not universally implemented.

Interestingly, Restier et al. [79] reported, using the Likert scale, that health professionals underrated the prevalence of malnutrition (Estimated: 16.8%; Calculated: 34.8%) and overrated frequency of assessment (Estimated: 80.6%; Measured: 43.1%) thereby jeopardizing the quality of care and patients’ safety. Along the same line, Huysentuyt et al. [49] in a Belgian nationwide questionnaire-based survey revealed that half of the pediatric departments in secondary-level hospitals did not perform nutritional assessment, the reasons given being lack of training (46.9%), unawareness of the Nutrition Support Team (42.2%) and lack of time (29.7%). Recently, De Longueville et al. [80] reported that, when compared to an in-house developed software (Evalnut) that not only evaluates risk of malnutrition but it also provides advice for its management, dieticians were the most aware of the importance of nutritional assessment and management during the hospitalization period. Last, a South Korean nationwide hospital-based survey [81] revealed that only half of the surveyed tertiary- and general-care hospitals had the required nutritional support staff, and that their knowledge was insufficient leading to a failing identification upon admission.

The French group PREDIRE [82] were the first to describe a protocol for a multifaceted randomized controlled intervention. Its aim was to improve the assessment and care of malnourished hospitalized children coordinated by a Nutritional Support Team composed of a pediatrician specialized in gastroenterology and nutrition, and 2 dieticians. Figure 2 illustrates the components of the intervention as well as the target groups and the stages of nutritional treatment. The objectives were: (1) To raise cognizance concerning the malnutrition issue; (2) Coach clinical teams regarding guidelines for good practice; (3) Facilitate screening of malnourished children through an electronic alert tool; (4) Assist in decision making regarding care and treatment of malnourished children; and (5) Coordinate nutritional care among the different health care professionals. In 2015 [83] the PREDIRE Study Group published the results of the planned intervention and showed that clinical practices for all outcomes measured were meaningfully perfected. They further reported that the investigation on the etiology and management of malnutrition were dramatically improved. This innovative initiative, based in part on an electronic system to detect malnutrition, resulted into a significant improvement of patient care.

## 8. Hospital Food Services

As hospital food service is often a low priority, the budget allocation for food is often limited. When public health budgets cuts are needed in the hospital, food items will often be targeted. Considering that patients consuming <50% of presented hospital meals are more likely to lose weight and have longer LOS, it is pertinent to promote the pleasure of eating and to maximize the overall food intake [42,84]. Limited menu food selection, child-unfriendly food, inflexible mealtimes and unfavorable meal delivery systems are all frequent barriers to poor nutrition in hospitalized patients [70,85,86]. Moreover, the hospital is an unfamiliar environment that can be intimidating and confusing for a sick child. Encouraging children to eat hospital meals can be a real challenge for parents and may result in conflict [87]. Many of hospitalized children will ask for comfort foods that are known to provide emotional comfort and security. A Canadian study conducted in an adult setting revealed that interruption of mealtimes and not receiving food when a meal was missed were obstacles to poor food intake in patients [88]. A pilot study conducted at the Toronto Sick Kids Hospital evaluating the impact of meal service type on the satisfaction and food intake of children concluded that a hotel room meal service model versus their current system of cold-plating tray delivery provided greater satisfaction and better food consumption while reducing costs and waste [89]. Williams et al. [85] have also studied the effect of better-need suited food service for hospitalized children by allowing children to call the kitchen at any time between 7:00 and 19:00 to have a meal delivered to their room. This strategy led to higher caloric and protein intake, 28% and 18% respectively, significant decrease in food wasting and potential cost savings of approximately $ 35,712/y. A cost reduction in the food service is often a compelling factor to encourage institutions reluctant to include organizational change of this magnitude. In short, the room service could be a win-win situation, being beneficial for the patient and the hospital food service department. Another Canadian pediatric study showed similar results, indicating that the satisfaction of the children had improved following the implementation of a “hotel room service model” [90]. Dieticians generally agree that modifying the hospital diet to the child’s taste and preference is a crucial point in improving recovery [91].

## 9. Conclusions

Our review emphasizes that child malnutrition in pediatric hospitals persists despite past recommendations and guidelines. The lack of consistency in the type of measures and their cut-off values prevents estimating the true prevalence of malnutrition. As reflected in the review, the assessment of nutritional status in hospitalized children is complex, many factors having to be considered when choosing indicators for documenting malnutrition. Health professionals should consider the strengths and limitations of indices, as some may lead to over- or under-estimation of malnutrition. Unfortunately, the hospital itself may have a potential negative impact in the nutrition of the child. Multiple practices have been acknowledged as unfavorable for the patient such as the absence of nutritional screening and assessment, unnecessary prolonged periods of fasting or poor flexibility with mealtimes. These organizational and logistic barriers undoubtedly result in increased complications, poorer tolerance to treatment and prolonged length of hospital stay for the pediatric inpatient. Future studies need to concentrate on improving nutrition screening or assessment tools by adding for example biological markers in the algorithms. Zhang et al. [92] have taken an initiative in that direction by performing a systematic review and meta-analysis for evaluating blood biomarkers associated with risk of malnutrition in elderly persons. Such a study is warranted in childhood. Investigations should also include evaluation of outcomes of nutritional intervention strategies tailored to pediatric care.

## Figures and Tables

**Figure 1 nutrients-11-00236-f001:**
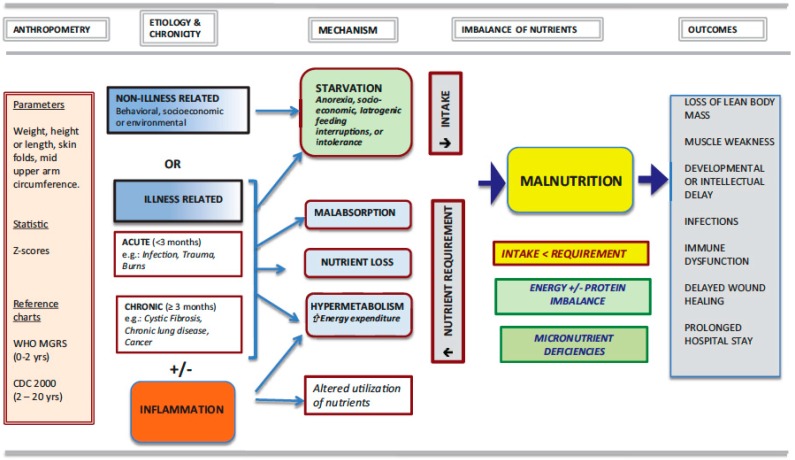
The key concepts in establishing malnutrition in hospitalized children. Reproduced with permission from Mehta NM, et al [31]. *Journal of parenteral and enteral nutrition*. 2013, 37, 460–481.

**Figure 2 nutrients-11-00236-f002:**
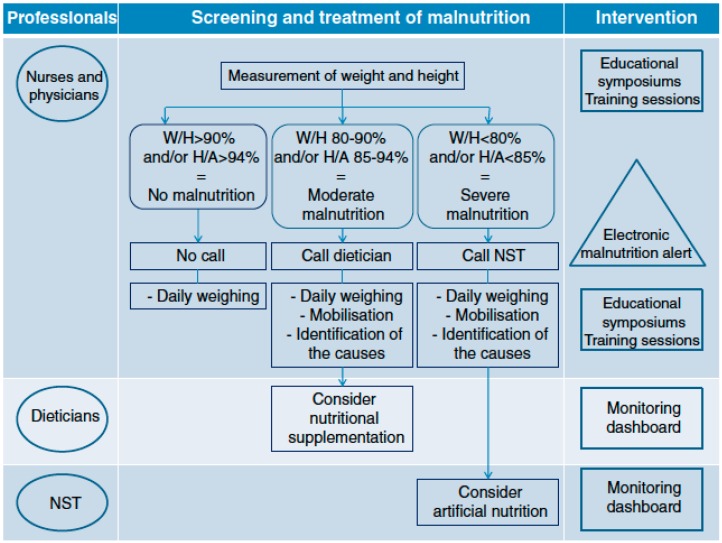
Components of the intervention as well as the target groups and the stages of nutritional treatment. Reproduced with permission according to the Creative Commons Public Domain Mark 1.0 Touzet S et al. [82]. BMC Health Services Research 2013.

**Table 1 nutrients-11-00236-t001:** Pediatric nutritional screening tools.

Tools	Anthropometric Evaluation	Nutritional Intake	Medical Condition	Others	Score	Ref
STAMP	Weight & height measurement	Nutritional intake	Pathology Medical condition		Score ≥ 4 = HNR	[40]
STRONG_kids_	Reported weight loss or no gain	Nutritional and Impaired intake	Pathology/high risk disease	Subjective clinical assessment (diminished fat &/or muscle mass &/or hollow face)	Score ≥ 3 = HNR	[41]
PNRS	Weight loss	Food intake <50%	Feeding interference Medical condition	Pain	Score ≥ 3 = HNR	[42]
PNST	Reported recent weight loss	Reported feeding in the last few weeks		Not Fully validated	Score ≥ 4 = HNR	[43]
PMST	Weight & height, BMI	Food intake	Pathology Medical condition		Score ≥ 4 = HNR	[44]
PYMS	Body Mass Index (BMI) <2% pertentile. (<−2 SD) on UK 1990 growth chart Weight loss	Changes in nutritional intake	Pathology Medical condition		Score ≥ 2 = HNR	[45]
PeDiSMART	WFA (z Score)	Nutritional intake	Disease impact	Computer/Not fully validated	Score ≥ 18 = HNR	[46]

BMI: Body Mass Index; GI: Gastrointestinal; HNR: High Nutritional Risk; PNRS: Pediatric Nutritional Risk Score; STRONG_kids_ Screening Tool for Risk On Nutritional status and Growth; PeDiSMART: Pediatric Digital Scaled Malnutrition Risk Screening Tool; PNST: Pediatric Nutrition Screening Tool; PYMS: Pediatric Yorkhill Malnutrition Screening; STAMP: Screening Tool for the Assessment of Malnutrition in Pediatrics; PMST: Pediatric Malnutrition Screening Tool.

**Table 2 nutrients-11-00236-t002:** Prevalence of undernutrition among children and adolescents admitted in pediatric hospitals.

Geographical Location	Population Studied	Clinical Setting	Screening Tools	Anthropometric Parameters	Prevalence	Ref
Belgium	0.8–17 y *N* = 379	Tertiary & secondary care facilities	No specific	WFH < −2 SD BMI < −2 SD %WFH < 80% MUAC < −2 SD Any one variable	9.0% 9.8% 2.4% 3.8% 13.5%	[14]
Canada	Birth–18 y *N* = 173	General pediatric unit	PNRS	Scoring	20.2%	[56]
Canada	1 m–18 y *N* = 307	Tertiary Pediatric Care Facilities	STRONG_kids_	Scoring WFA < −2 SD HFA < −2 SD WFH or BMI < −2 SD Any one variable	26.6% 10.4% 14.0% 9.1% 19.5%	[59]
Europe	1 m–18 y *N* = 1258	14 Hospital Centres General pediatric wards & pediatric surgery	PYMS STAMP STRONG_kids_	Scoring	22% 22% 10%	[60]
France	1–≥72 m *N* = 296	Tertiary care facility	PNRS	PIBW < 85%	26%	[42]
France	>6 m *N* = 52	Tertiary care facility	NRS	BMI < −2 SD Scoring	12% 26%	[55]
France	2 m–16 y *N* = 280	Tertiary care facility	No specific	WFH < −2 SD	11%	[58]
France	1 d–16 y *N* = 923	Primary & Tertiary Care Facilities	No specific	WFH < −2 SD	11.9%	[61]
Germany	7.9 ± 5 y *N* = 475	Tertiary care facility	Waterlow classification	Median WFH < 80% TST < 10% Perc.	6.1% 17.2%	[52]
Italy	1 m–20 y *N* = 1994	Tertiary care & General pediatric wards	No specific	BMI or WFH<−2 SD	13.2%	[62]
Turkey	1 m–23 y *N* = 528	General pediatric unit	No specific	WFA < −2 SDWFH < −2 SDBMI < −2 SD	36.6% 27.7% 7.4%	[57]
Turkey	1 m–18 y *N* = 1513	Nationwide hospitals	PYMS *N* = 919 STRONG_kids_	Scoring BMI < −2 SD WFA < −2 SD HFA < −2 SD Scoring	39.7% 9.5% 14.8% 16.2% 3.6%	[63]
UK	0.6–16 y *N* = 226	Tertiary care facility	No specific	WFA < −2 SD or <5% Perc. HFA < −2 SD or <5% Perc. WFH: <80% of STD	8% 11% 16%	[54]
US	<2–18 y *N* = 268	Tertiary care facilities	Waterlow classification	Median WFH < 80%	7.1%	[51]
US	<24 m *N* = 121	Cardiac intensive tertiary care facility	Waterlow classification	Median WFH < 80%	17.4%	[53]
US	2–18 y *N* = 1747	Tertiary care facility	No specific	BMI ≤ 5% Perc.	24.5%	[64]
US	1 m–17 y *N* = 2.14 × 10^6^	Nationwide hospitals	No specific	% discharges	2.6%	[65]

BMI: body mass index; WFH: weight for height (Waterlow classification, evaluation of acute protein-energy malnutrition, wasting); WFA: weight for age (acute, underweight); HFA: height for age (chronic stunting); Perc.: Percentile; PIBW: % of ideal body weight; MUAC: mid-upper arm circumference; TST: Triceps skinfold thickness; STD: standard deviation, d: day; m: month; y: year.

**Table 3 nutrients-11-00236-t003:** Hospital practices that may worsen the nutritional status of hospitalized paediatric patients.

Failure to document the patient’s weight and height and to plot these measurements on appropriate growth charts; Improper growth charts use; Inaccurate anthropometric measurements/lack of adequate equipment; Failure to document poor nutritional status in the hospital charts/lack of dietetic referral; Inadequate nutritional intake due to medical procedures/hospital food; Failure to prioritize nutrition care; Lack of nutritional screening on admission and inpatient monitoring during the hospital stay; Inadequate nutritional education and training of hospital staff;

Adapted from [69,71,72].

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
