# Peer review of "Prevalence of Malnutrition in Pediatric Hospitals in Developed and In-Transition Countries: The Impact of Hospital Practices"

_nutrients, 2019, doi:10.3390/nu11020236_

Round 1
Reviewer 1 Report
This is a nice review of an important and understudied topic. I hope that this manuscript will spur renewed attention to this topic in high income countries. I have few recommendations and questions which may help the authors clarify certain aspects of the paper:
1) There needs to be clarification of how the authors define “malnutrition” very early in the paper. It becomes clear that you’re primarily talking about acute malnutrition and chronic malnutrition, what may be considered wasting and stunting. But some readers would assume malnutrition would include micronutrient deficits, or even obesity as part of the double burden of malnutrition. Additionally, it would helpful to know if the authors differentiate “failure to thrive” from malnutrition.
2) The sentence beginning on line 34 and ending on line 37, is s a bit confusing and could be clarified. The point is correct, low-income setting is multifactorial, high-income settings are predominantly disease related, but could clearer.
3) You drew data from “industrialized” countries. How was this defined? You discuss papers from Turkey and China, where I imagine there are variety of settings, some of which are comparable western hospitals and other which are more comparable middle-income settings. Can you comment on the difference between evidence from North America and Europe to the rest of the world?
4) Through out the document you refer to the prevalence of malnutrition. You do mention that the studies you included were not community based, i.e. they looked at inpatient children, or children presenting to care. Please make it clear that the prevalences you refer to are among children seeking care (or in care) and not among the general population. This is particularly important for the abstract and conclusion. It would also be worth including data about the community prevalence in the introduction, and mentioning that the prevalence among inpatients will be higher as malnourished children are likely to have of develop medical complications.
This is nice contribution. Good luck with your submission.
Author Response
This is a nice review of an important and understudied topic. I hope that this manuscript will spur renewed attention to this topic in high-income countries. I have few recommendations and questions that may help the authors clarify certain aspects of the paper:
1) There needs to be clarification of how the authors define “malnutrition” very early in the paper. It becomes clear that you’re primarily talking about acute malnutrition and chronic malnutrition, what may be considered wasting and stunting. But some readers would assume malnutrition would include micronutrient deficits, or even obesity as part of the double burden of malnutrition. Additionally, it would helpful to know if the authors differentiate “failure to thrive” from malnutrition.
Response: Methods section under Malnutrition: A broad concept. We have added the following sentence: The generic term malnutrition encompasses deficient, excessive or imbalanced intake of a variety of nutrients jeopardizing the health status. It can be causal or consequential. The present review addresses acute and chronic undernutrition, the latter including wasting and stunting.
2) The sentence beginning on line 34 and ending on line 37, is s a bit confusing and could be clarified. The point is correct, low-income setting is multifactorial, high-income settings are predominantly disease related, but could clearer.
Introduction: We have modified the sentence to read: Independently of the income stetting, malnutrition is multifactorial. Whereas in low-income countries, malnutrition is often, but not solely, attributable to limited access to food and/or medical care, in developed and in-transition countries it is often triggered by disease.
3) You drew data from “industrialized” countries. How was this defined? You discuss papers from Turkey and China, where I imagine there are variety of settings, some of which are comparable western hospitals and others that are more comparable middle-income settings. Can you comment on the difference between evidence from North America and Europe to the rest of the world?
Response: Introduction. We have added the following sentence: We have selected examples of countries that are classified developed economies and economies in transition according to the definition used by World Economic Situation and Prospects (WESP) and prepared by the Development Policy and analysis Division (DPAD) of the Department of Economic and Social Affairs of the United Nations Secretariat (UN/DESA)http://www.un.org/en/development/desa/policy/wesp/wesp_current/2014wesp_country_classification.pdf.
Response: Title. The title has been modified to reflect this. It now reads: Prevalence of malnutrition in pediatric hospitals in developed and in-transition countries and the impact of hospital practices: stepping inside the hospital walls.
Throughout the text we have also replaced industrialized countries by “developed and in-transition countries”
4) Throughout the document you refer to the prevalence of malnutrition. You do mention that the studies you included were not community based, i.e. they looked at inpatient children, or children presenting to care. Please make it clear that the prevalence you refer to are among children seeking care (or in care) and not among the general population. This is particularly important for the abstract and conclusion. It would also be worth including data about the community prevalence in the introduction, and mentioning that the prevalence among inpatients will be higher as malnourished children are likely to have of develop medical complications.
Response: Introduction. In order to make clear that the prevalence we refer to are among children seeking care (or in care) and not among the general population, we have added the following in the Introduction: The objectives of the present review is to provide a short historical account, briefly describe the tools developed for evaluating malnutrition, present an overview on the prevalence of pediatric malnutrition in patients seeking care and/or hospitalized, and document the compliance of the published guidelines in pediatric hospitals in developed and in-transition countries.
Response: Introduction. We also included a reference to the burden of protein-energy malnutrition on the rate of death to illustrate its impact in different economical settings. It reads: Of importance, the report of the Global Burden of Disease Study 2013 (GBD) revealed that protein-energy malnutrition accounted globally for 9.8/100,000 age-standardized deaths in the largest 50 countries for child and adolescent populations with ages ranging from 0-19 years. More alarming, when classifying the data according to the level of development, it accounted for 11/100,000 age-standardized deaths in the developing countries and 0.1/100,000 age-standardized deaths in developed countries.
Response: Abstract. We have modified sentences in the abstract to read: The reported prevalence of undernutrition in pediatric patients seeking care or hospitalized varies considerably, ranging from 2.5 to 51%. This disparity is mostly due to the diversity of the origin of populations studied, methods used to detect and assess nutritional deficiencies, as well as the lack of consensus for defining pediatric undernutrition. Last, the prevalence among inpatients is likely to be higher than that observed for the community at large since malnourished children are likely to have of develop medical complications.
This is nice contribution. Good luck with your submission
We thank the reviewer for his encouragement.
Reviewer 2 Report
The authors are to be commended on their very extensive review on the literature about disease related malnutrition in hospitalized children.
It is however not clear for me what the added value of this review is above the currently existing narrative and systematic reviews on the topic. This paper would benefit from a section 'what is already known - what this study adds'.
The authors have provided us with a very extensive review of the literature, but this has made the paper very long. There is a lot of overlap between the text and the tables. I would recommend for example in the section on nutrition screening tools to limit the text to a critical appraisal of each tool and to refer to the table for additional info. Also, the SGNA is not intended as a screening tool, but as an assessment tool and should be removed from this section. There is also a recent screening tool about nutrition risk in infants that I did not find in this overview.
I would also recommend to add some sentences explaining the difference between nutrition screening and assessment, which is confused often in literature, and even in this paper sometimes.
the section on prevalence of acute malnutrition shows again much overlap between text and table, so I recommend to shorten the text. A critical appraisal of the definitions used for acute malnutrition is missing here. Do the authors agree with all measures that have been used ?
There is more recent literature about barriers for nutritional assessment than the one used in Table 3, suggest to revise this.
Author Response
The authors are to be commended on their very extensive review on the literature about disease related malnutrition in hospitalized children.
It is however not clear for me what the added value of this review is above the currently existing narrative and systematic reviews on the topic. This paper would benefit from a section 'what is already known - what this study adds.
First of all, we are very excited that the Reviewer has found our review exhaustive as many previous articles remained fragmented and dealt with only a few aspects at a time. Our review emphasizes in particular that despite past recommendations, child malnutrition remains a persisting and current in pediatric hospitals. In addition, our review has repositioned the assessment tools for malnutrition.
The authors have provided us with a very extensive review of the literature, but this has made the paper very long. There is a lot of overlap between the text and the tables. I would recommend for example in the section on nutrition screening tools to limit the text to a critical appraisal of each tool and to refer to the table for additional info. Also, the SGNA is not intended as a screening tool, but as an assessment tool and should be removed from this section. There is also a recent screening tool about nutrition risk in infants that I did not find in this overview.
I would also recommend adding some sentences explaining the difference between nutrition screening and assessment, which is confused often in literature, and even in this paper sometimes.
Response. We realize that we have interchangeably used screening and assessing when referring to the various tools, hence causing a little confusion for the reader. In essence, the tools that have been developed since 1995 were meant to assess the nutritional status of various groups. Since then, in the present context, the term screening refers to the process of examining groups to distinguish more clearly well-nourished from malnourished persons or those at high risk based on the nutritional assessment tools.
The section on prevalence of acute malnutrition shows again much overlap between text and table, so I recommend shortening the text.
A critical appraisal of the definitions used for acute malnutrition is missing here.
Response. SUCCINCT HISTORICAL REFERENCE section. We have modified the sentence “Waterlow et al. (31) proposed in 1972 that acute malnutrition should be defined independently of age and they suggested using weight in relation to height. They further differentiated wasting from stunting; wasting or acute malnutrition being described by the weight-for-height (WFH) percentile and stunting or chronic malnutrition by the height-for-age (HFA) percentile.
Do the authors agree with all measures that have been used?
Response. Conclusion. We believe this point has been addressed in the Conclusion section. However, we agree that it could be better expressed. Therefore, we have now modified the text to read “Malnutrition remains at the present time a common issue in the pediatric setting. The lack of consistency in the type of measures and their cut-off values prevents estimating the true prevalence of malnutrition. As reflected in this review, the evaluation of nutritional status in hospitalized children is complex since many factors have to be considered when choosing indicators for malnutrition Documentation. Health professionals should consider the strengths and limitations of indices, as some may lead to over- or under-estimation of malnutrition. Unfortunately, the hospital itself may have a potential negative impact on the nutrition of the child. Multiple practices have been acknowledged as unfavorable for the patient such as the absence of nutritional assessment, unnecessary prolonged periods of fasting or poor flexibility with mealtimes. These organizational and logistic barriers undoubtedly result in increased complications, poorer tolerance to treatment and prolonged length of hospital stay for the pediatric inpatient. Future studies need to concentrate on improving nutrition assessment tools by adding for example biological markers in the algorithms. They should also include evaluation of outcomes of nutritional intervention strategies tailored to pediatric care.”
There is more recent literature about barriers for nutritional assessment than the one used in Table 3. I suggest revising this.
We have updated the references.